# UNDERSTANDING YOUR AGENT: LEVERAGING LARGE LANGUAGE MODELS FOR BEHAVIOR EXPLANATION

## ABSTRACT

Intelligent agents such as robots are increasingly deployed in real-world, safety-critical settings. It is vital that these agents are able to explain the reasoning behind their decisions to human counterparts; however, their behavior is often produced by uninterpretable models such as deep neural networks. We propose an approach to generate natural language explanations for an agent's behavior based only on observations of states and actions, thus making our method independent from the underlying model's representation. For such models, we first learn a behavior representation and subsequently use it to produce plausible explanations with minimal hallucination while affording user interaction with a pre-trained large language model. We evaluate our method in a multi-agent search-and-rescue environment and demonstrate the effectiveness of our explanations for agents executing various behaviors. Through user studies and empirical experiments, we show that our approach generates explanations as helpful as those produced by a human domain expert while enabling beneficial interactions such as clarification and counterfactual queries.

## 1 INTRODUCTION

Rapid advances in artificial intelligence and machine learning have led to an increase in the deployment of robots and other embodied agents in real-world, safety-critical settings (Sun et al., 2020; Fatima & Pasha, 2017; Li et al., 2023). As such, it is vital that practitioners – who may be laypeople that lack domain expertise or knowledge of machine learning – are able to query such agents for explanations regarding *why* a particular prediction has been made – broadly referred to as explainable AI (Amir et al., 2019; Wells & Bednarz; Gunning et al., 2019). While progress has been made in this area, prior works tend to focus on explaining agent behavior in terms of rules (Johnson, 1994), vision-based cues (Cruz & Igarashi, 2021; Mishra et al., 2022), semantic concepts (Zabounidis et al., 2023), or trajectories (Guo et al., 2021). However, it has been shown that laypeople benefit from natural language explanations (Mariotti et al., 2020; Alonso et al., 2017) since they do not require specialized knowledge to understand (Wang et al., 2019), leverage human affinity for verbal communication, and increase trust under uncertainty (Gkatzia et al., 2016).

In this work, **we seek to develop a framework to generate natural language explanations of an agent's behavior given only observations of states and actions**. By assuming access to only behavioral observations, we are able to explain behavior produced by *any* agent policy, including deep neural networks (DNNs). Unlike prior methods, which exhibit limited expressivity due to utilizing language templates (Hayes & Shah, 2017; Kasenberg et al., 2019; Wang et al., 2019) or assume access to a large dataset of human-generated explanations (Ehsan et al., 2019; Liu et al., 2023), we propose an approach in which large language models (LLMs) can be used to generate free-form natural language explanations in a few-shot manner. While LLMs have shown considerable zero-shot task performance and are well-suited to generating natural language explanations (Wiegreffe et al., 2021; Marasović et al., 2021; Li et al., 2022), they are typically applied to commonsense reasoning as opposed to explaining model behavior and are prone to hallucination – a well-known phenomenon in which false information is presented as fact (McKenna et al., 2023). It is an open question as to how LLMs can be conditioned on an agent's behavior in order to generate plausible explanations

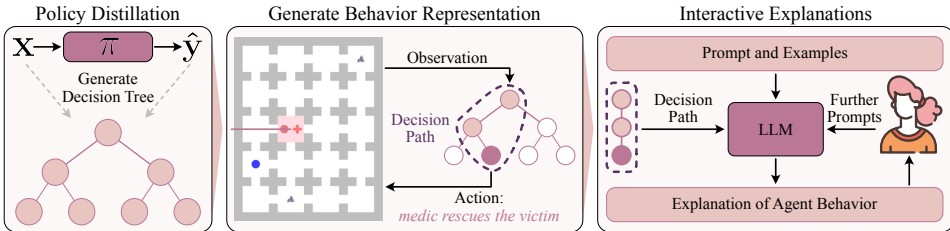

Figure 1: Overview of our three-step pipeline to explain policy actions: left: A black-box policy is distilled into a decision tree; middle: a decision path is extracted from the tree for a given state which contains a set of decision rules used to derive the associated action; right: we utilize an LLM to generate an easily understandable natural language explanation given the decision path. Lastly, a user can ask further clarification questions in an interactive manner.

while avoiding such hallucinations. We find this is a particularly important aspect as laypeople tend to struggle to identify hallucinated facts, as we observe in our participant studies in Sec. 4.3.

Our solution, and core algorithmic contribution, is the introduction of a *behavior representation* (BR), in which we distill an agent's policy into a locally interpretable model that can be directly injected into a text prompt and reasoned with, without requiring fine-tuning. A behavior representation acts as a compact representation of an agent's behavior around a specific state and indicates what features the agent considers important when making a decision. We show that by constraining an LLM to reason about agent behavior in terms of a behavior representation, we are able to greatly reduce hallucination compared to alternative approaches while generating informative and plausible explanations. An additional benefit of our approach is that it enables *interactive* explanations; that is, the user can issue follow-up queries such as clarification or counterfactual questions. This is particularly valuable, as explanations are social interactions conditioned on a person's own beliefs and knowledge (Miller, 2019) and thus, are highly individual and may require additional clarification to be comprehensible and convincing (Kass et al., 1988).

Our approach is a three-stage process (see Figure 1) in which we, 1) distill an agent policy into a decision tree, 2) extract a decision path from the tree for a given state which serves as our local *behavior representation*, and 3) transform the decision path into a textual representation and inject it into pre-trained LLM via in-context learning (Brown et al., 2020) to produce a natural language explanation. In this work we show how our framework can be applied to multi-agent reinforcement learning (MARL) policies – a particularly relevant setting given the complex dynamics and decision-making resulting from agent-agent interactions. Through a series of participant studies, we show that a) our approach generates model-agnostic explanations that laypeople significantly prefer over baseline methods and are preferred at least as much as those generated by a human domain expert; b) when an agent policy does not align with participant assumptions, participants find the ability to interact with our explanations helpful and beneficial; and c) our approach yields explanations with significantly fewer hallucinations than alternative methods of encoding agent behavior.

## 2 RELATED WORK

**Explainable Agent Policies**: Many works attempt to explain agent behavior through the use of a simplified but interpretable model that closely mimics the original policy (Puiutta & Veith, 2020; Verma et al., 2018; Liu et al., 2019; Shu et al., 2017), a technique which has long been studied in the field of supervised learning (Ribeiro et al., 2016). Although approaches that directly utilize inherently interpretable models with limited complexity during the training phase (Du et al., 2019) exist, many researchers avoid sacrificing model accuracy for interpretability. In this work, we follow an approach similar to (Guo et al., 2023), in which we leverage a distilled interpretable model to gain insight into how the agent's policy reasons.

**Natural Language Explanations**: Outside of explaining agent behavior, natural language explanations have received considerable attention in natural language processing areas such as commonsense reasoning (Marasović et al., 2020; Rajani et al., 2019) and natural language inference (Prasad et al., 2021). Unlike our setting in which we desire to explain a given *model's be-*

*havior*, these methods attempt to produce an explanation purely with respect to the given input and domain knowledge, e.g., whether a given premise supports a hypothesis in the case of natural language inference (Camburu et al., 2018). Although self-explaining models (Marasović et al., 2021; maj, 2022; Hu & Clune, 2023) are conceptually similar to our goal, we desire a model-agnostic approach with respect to the agent's policy and thus seek to explain the agent's behavior with a separate model. While recent works have investigated the usage of LLMs in explaining another model's behavior by reasoning directly over the latent representation (Bills et al., 2023), this approach has yielded limited success thus far and motivates our usage of an intermediate behavior representation.

## 3 LANGUAGE EXPLANATIONS FOR AGENT BEHAVIOR

We introduce a framework for generating natural language explanations for an agent from *only* observations of states and actions. Our approach consists of three steps: 1) we distill the agent's policy into a decision tree, 2) we generate a behavior representation from the decision tree, and 3) we query an LLM for an explanation given the behavior representation. We note that step 1 only needs to be performed once for a particular agent, while steps 2 and 3 are performed each time an explanation is requested. We make no assumptions about the agent's underlying policy such that our method is model agnostic; explanations can be generated for any model for which we can sample trajectories.

**Notation**: We consider an infinite-horizon discounted Markov Decision Process (MDP) in which an agent observes environment state $s_t$ at discrete timestep $t$, performs action $a_t$, and receives the next state $s_{t+1}$ and reward $r_{t+1}$ from the environment. The MDP consists of a tuple $(\mathcal{S}, \mathcal{A}, R, T, \gamma)$ where $\mathcal{S}$ is the set of states, $\mathcal{A}$ is the set of agent actions, $R : \mathcal{S} \times \mathcal{S} \to \mathbb{R}$ is the reward function, $T : \mathcal{S} \times \mathcal{A} \times \mathcal{S} \to [0, 1]$ is the state transition probability, and $\gamma \in [0, 1)$ is the discount factor. As in standard imitation learning settings, we assume the reward function $R$ is unknown and that we only have access to states and actions sampled from a stochastic agent policy $\pi^*(a|s) : \mathcal{A} \times \mathcal{S} \to [0, 1]$.

### 3.1 DISTILLING A DECISION TREE

Our first step is to distill the agent's underlying policy into a decision tree, which acts as an interpretable *surrogate*. The decision tree is intended to faithfully replicate the agent's policy while being interpretable, such that we can extract a behavior representation from it. Given an agent policy $\pi^*$, we distill a decision tree policy $\hat{\pi}$ using the DAgger (Ross et al., 2011) imitation learning algorithm which minimizes the expected loss to the agent's policy under an induced distribution of states,

$$\hat{\pi} = \arg \min_{\pi \in \Pi} \mathbb{E}_{s^*, a^* \sim \pi^*}[\mathcal{L}(s^*, a^*, \pi)], \tag{1}$$

for a restricted policy class $\Pi$ and loss function $\mathcal{L}$. This method performs iterative data aggregation consisting of states sampled from the agent's policy and the distilled decision tree, in order to overcome error accumulation caused by the violation of the i.i.d. assumption. While decision trees are simpler than other methods such as DNNs, it has been shown that they are still capable of learning reasonably complex policies (Bastani et al., 2018). Intuitively, DNNs often achieve state-of-the-art performance not because their representational capacity is larger than other models, but because they are easier to regularize and thus train (Ba & Caruana, 2014). However, distillation is a technique that can be leveraged to distill the knowledge contained within a DNN into a more interpretable decision tree (Hinton et al., 2015; Frosst & Hinton, 2017; Bastani et al., 2017).

### 3.2 BEHAVIOR REPRESENTATION GENERATION

The distilled policy $\hat{\pi}$ consists of a set of decision rules which approximate the decision-making process of the agent's policy $\pi^*$. Given a state $s_t$ and action $a_t$ taken by the agent, we extract a decision path $dp = Path(\hat{\pi}, s_t)$ which acts as a *locally* interpretable model of the agent's behavior. The path $dp$ consists of a subset of the decision rules in $\hat{\pi}$ which produce the action $a_t$ in state $s_t$, and is obtained by simply traversing the tree from root to leaf. These decision rules approximate the agent's underlying decision-making rationale in state $s_t$ and can be used to infer intent.

Figure 2 shows example decision paths for agents operating in an Urban Search and Rescue (USAR) environment where heterogeneous agents with different action spaces learn to coordinate to rescue

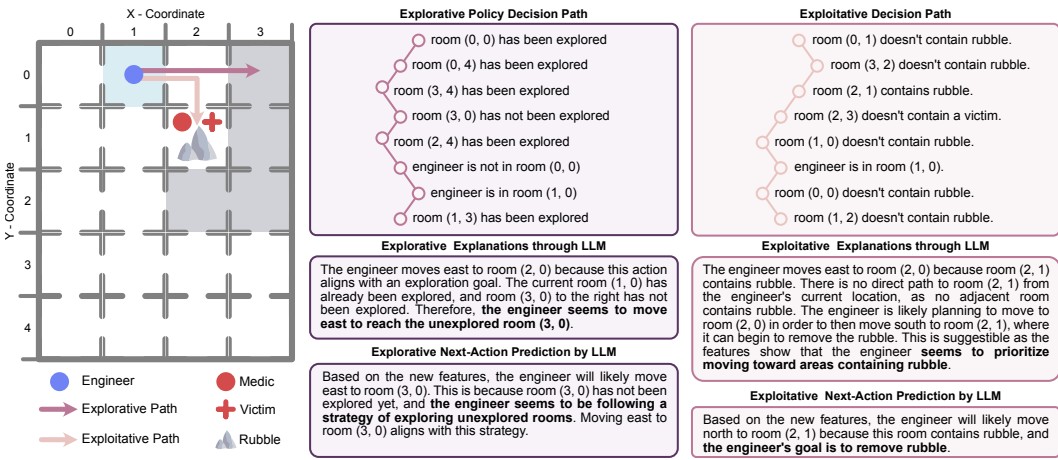

Figure 2: An example of am ambiguous state, in which the engineer's current state can be induced by following two distinct behaviors: *Exploit*, which prioritizes removing rubble as soon as possible, and *Explore*, which prioritizes visiting unexplored rooms. Given the current state (engineer at (1, 0)) and intended action (going east), their follow-up action is ambiguous depending on which behavior is utilized by the engineer (*Explore*: Purple; *Exploit*: Pink). The corresponding decision paths are shown for each possible behavior and the resulting natural language explanations after transforming into a behavior representation.

victims Lewis et al. (2019); Freeman et al. (2021). We adopt the environment of Guo et al. (2023) in which there are two agents: a *Medic*, responsible for healing victims, and an *Engineer* responsible for clearing rubble. The left decision path, denoted as *Explore* Decision Path, corresponds to an agent exhibiting exploration behavior, i.e. it fully explores the environment before removing any pieces of rubble. The right decision path, *Exploit* Decision Path, corresponds to an exploitative agent which greedily removes rubble as it is discovered. We can observe how these different behaviors are reflected in their respective decision paths – the *Explore* path largely consists of decision rules examining whether rooms have been explored, while the *Exploit* path consists of rules checking for the existence of rubble. This enables effective reasoning, e.g. because the *Explore* agent *only* checks for explored rooms before taking its action, we can infer that the agent is currently only interested in exploration and is choosing to ignore any visible rubble.

We refer to such a decision path as a behavior representation, and it serves as a compact encoding of the agent's behavior. This representation is effective for three reasons: a) decision tree depth is usually constrained in order to prevent overfitting, which means even complex policies can yield decision paths that fit into relatively small LLM context windows; b) decision paths can be readily translated into natural language via algorithmic templates and injected into an LLM prompt – meaning no fine-tuning is required (Brown et al., 2020), which is an important factor given the lack of human-annotated explanations for agent behavior; and c) they provide an intuitive and explicit ordered set of rules with which an LLM can reason, resulting in more effective explanations and reduced hallucination compared to alternative behavior encodings. We show in Sec. 4 that our proposed behavior representation strongly outperforms alternative encodings.

### 3.3 IN-CONTEXT LEARNING WITH BEHAVIOR REPRESENTATIONS

The last step in our approach is to define a prompt that constrains the LLM to reason about agent behavior with respect to a given behavior representation. Our prompt consists of four parts: a) a concise description of the environment the agent is operating in, e.g., state and action descriptions, b) a description of what information the behavior representation conveys, c) in-context learning examples, and d) the behavior representation and action that we wish to explain. An example of this prompt is shown in our appendix (See. A). All parts except for (d) are pre-defined ahead of time and remain constant for all queries, while our framework provides a mechanism for automatically constructing (d). Thus, our system can be queried for explanations with no input required by the user unless they wish to interact and submit follow-up queries.

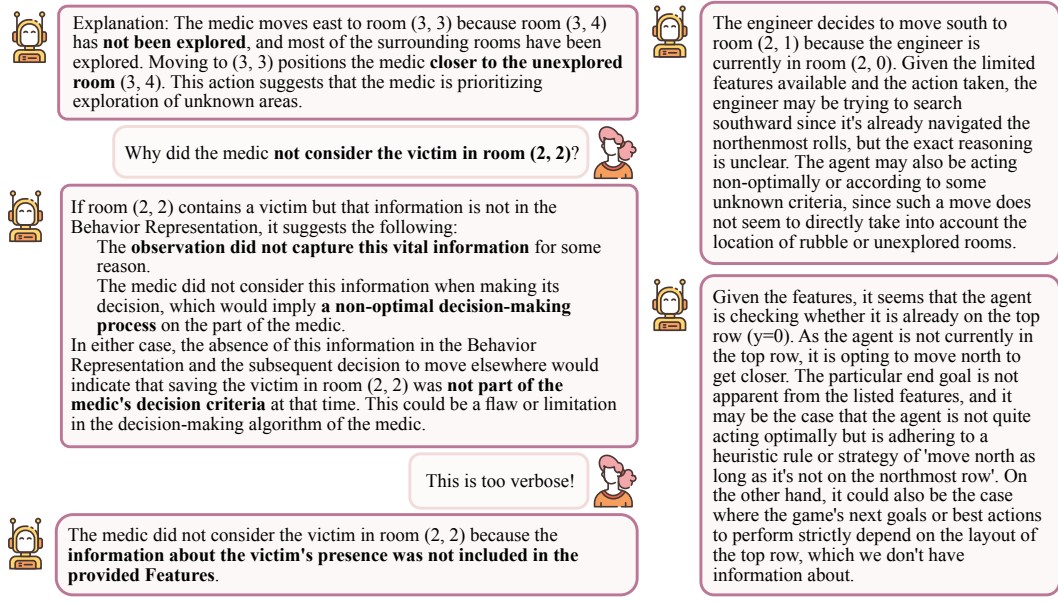

Figure 3: Left: Interactive Conversation: The user is asking more clarification questions about the initial explanation generated by our method. Right: Explanation of a *Fixed* Agent: the LLM is able to detect and rationale about suboptimality.

The ability to ask such follow-up questions plays a crucial role when interacting with laypeople, who may prefer additional explanations of an agent's behavior. Utilizing an LLM to generate our explanations allows a user to ask such questions to, for example, request further clarification, "what-if" scenarios (counterfactual), further summaries, likely next actions, or other requests. As we show in Sec. 5.2, this is particularly valuable when the agent's actions are not aligned with the user's expectations. In our experiments, we have observed two main types of questions: a) requests for further clarification, e.g. *"Why did the agent not consider feature X when making its decision?"*, and 2) counterfactual questions, e.g., "What if feature Y were present instead of feature X?" See section 5.2 for further details.

# 4 QUANTITATIVE RESULTS AND ANALYSIS

We quantitatively evaluate the performance of our proposed approach in a simulated multi-agent Urban Search and Rescue task with the goal of answering the following questions: **1)** Does our behavior representation enable the LLM to reason about varying behaviors and identify the underlying decision-making rationale? **2)** Does our behavior representation enable the LLM to infer *future* behavior? **3)** How is hallucination affected by our choice of behavior representation?

**Experimental Setup** Our experimental setting is a partially observable Urban Search and Rescue task in which two agents work cooperatively in order to rescue victims and remove rubble. We model this task as a 2D Gridworld consisting of 20 rooms arranged in a $5 \times 4$ grid. Both agents can navigate to adjacent rooms in the environment and each have role-specific actions – the engineer can remove rubble which may be hiding victims, and the medic can rescue victims after rubble has been removed. This environment is partially observable and the agents must first traverse the environment in order to locate both rubble and victims. The agents can exhibit one of three possible behaviors:

- **Explore**: Rubble and victims are immediately removed or rescued upon discovery.
- **Exploit**: Agents explore all rooms before backtracking to remove rubble/rescue victims.
- **Fixed**: Agents ignore rubble and victims and simply move in a pre-determined pattern.

We generate natural language explanations for each agent's behavior using one of three methods:

- **BR (Path)**: Our proposed method which uses a decision path as a behavior representation.

| | Method | Long-term | | | Short-term | | | Ambiguous | | |
|---|---|---|---|---|---|---|---|---|---|---|
| | | Strategy | Category | Goal | Strategy | Category | Goal | Strategy | Category | Goal |
| Exploit | **BR (Path)** | 0.70 | 0.75 | **0.75** | **1.00** | **1.00** | **1.00** | **0.90** | **0.85** | **0.85** |
| | BR (States) | **0.75** | **0.80** | 0.75 | 0.75 | 0.75 | 0.75 | 0.60 | 0.75 | 0.75 |
| | No BR | 0.25 | 0.25 | 0.25 | 0.90 | 0.95 | 0.95 | 0.70 | 0.75 | 0.75 |
| Explore | **BR (Path)** | **0.90** | **0.75** | **0.25** | **1.00** | **1.00** | 0.70 | **0.90** | **0.80** | **0.35** |
| | BR (States) | 0.40 | 0.05 | 0.05 | 0.70 | 0.95 | **0.95** | 0.00 | 0.00 | 0.00 |
| | No BR | 0.20 | 0.30 | 0.15 | 0.40 | 0.90 | 0.90 | 0.00 | 0.05 | 0.00 |

Table 1: Explanation accuracy for randomly sampled states in which the agent is pursuing a long-term goal, short-term goal, or ambiguous goal while operating under two different behavior strategies: *Explore* and *Exploit*. All metrics represent accuracy (higher is better). Each value is computed over 20 samples: 10 each from medic and engineer. The best method in each column is bolded.

- **BR (States)**: An alternative behavior representation that uses a set of state-action pairs sampled from the agent's policy rather than a decision path.
- **No BR**: No behavior representation is given. This serves as a baseline to evaluate how well the LLM can reason about an agent given only an observation and no prior knowledge.

The generated behavior explanations are hand-annotated with regards to the following metrics:

- **Strategy**: Whether the agent's behavior (defined above) was identified in the explanation.
- **Category**: Whether the agent's goal category was identified in the explanation, e.g., the agent is moving towards a rubble/victim/unexplored room.
- **Goal**: Whether the agent's specific goal was identified in the explanation, e.g., the agent is moving to rubble in room (1, 2).
- **Action**: Whether the agent's next action was successfully predicted.
- **Intent**: Whether the agent's intent for taking the next action was successfully identified, e.g., the agent moved to room (1, 1) because it will be closer to rubble in room (1, 2).

Unlike works which evaluate natural language explanations in domains such as natural language inference, to the best of our knowledge there are no datasets consisting of high-quality explanations generated over agent behavior. Due to the time and effort required to construct such datasets, ours is relatively small in comparison and precludes the usage of automatic metrics such as BLEU, which work well only over large corpuses.

## 4.1 EVALUATING EXPLANATION QUALITY

We evaluate explanation quality by generating explanations for state-action pairs randomly sampled from agent trajectories produced by each behavior type. States are grouped into three categories: **Long-term** — The agent is moving to a room/rubble/victim but won't get there in the next time step; **Short-term** — in which agent is pursuing a short-term goal, meaning it will reach the desired room/remove rubble/rescue victim in the next time step; and **Ambiguous** — where the *current* state-action can be induced by either exploration or exploitation behaviors, but the *next* state will yield different actions from each behavior. The results for each behavior are shown in Table 1 and *Fixed* in Table 3. We make the following observations.

**Explanations produced with BR (Path) are more accurate**. Explanations generated using a decision path behavior representation more accurately identify the agent's Strategy, Category, and Goal in every category except for Long-term *Exploit* when compared to other methods. We conjecture that the slightly reduced accuracy for long-term goals under exploitative behavior is due to the additional complexity associated with *Exploit* decision paths; they must simultaneously check for the presence of unexplored rooms and rubble, while the *Explore* decision paths do this sequentially, i.e., they first check all rooms' exploration status and *then* check for the presence of rubble.

**The LLM makes assumptions over expected behaviors**. The ambiguous states reveal an important insight: the LLM tends to assume an agent will act *exploitatively* when presented with an observation and a task description. We can see that all methods, including BR (States) and No BR, yield

|          | Method     | Long-term |        | Short-term |        | Ambiguous |        |
|----------|------------|-----------|--------|------------|--------|-----------|--------|
|          |            | Action    | Intent | Action     | Intent | Action    | Intent |
| Exploit  | **BR (Path)** | **0.80** | **0.75** | 0.40 | **0.75** | **0.85** | **0.85** |
| Exploit  | BR (States) | 0.65     | 0.55   | **0.75**   | **0.75** | 0.80    | 0.70   |
| Exploit  | No BR       | 0.55     | 0.50   | 0.65       | 0.65   | 0.80      | 0.75   |
| Explore  | **BR (Path)** | **0.95** | **1.00** | **0.80** | **0.95** | **0.90** | **0.95** |
| Explore  | BR (States) | 0.60     | 0.40   | 0.45       | 0.45   | 0.05      | 0.05   |
| Explore  | No BR       | 0.65     | 0.25   | 0.50       | 0.35   | 0.30      | 0.10   |

Table 2: Action prediction accuracy for randomly sampled states in which the agent is pursuing a long-term goal, short-term goal, or ambiguous goal. Action indicates whether the next action was correctly identified, while intent indicates whether the reason *why* the action was taken was identified. Each value is computed over 20 samples (10 for engineer and medic each)

relatively high accuracy when generating explanations for ambiguous state-actions under exploitative behavior. However, when the agent acts exploratory, BR (Path) continues to perform well (90% accuracy) while the other methods fail to get *any* explanations correct (0% accuracy). We find that this is because when presented with am ambiguous state that could be caused by multiple behaviors, the LLM assumes the agent acts exploitatively and only the decision path behavior representation is able to enforce a strong enough prior over agent behavior for correct reasoning. A similar trend can be observed with states sampled from the *Fixed* behavior in Table 3. BR (Path) yields an impressive 80% accuracy in detecting the fact that the agent is ignoring victims and rubble and pursuing a pre-determined path, yet again the LLM assumes exploitative behavior for BR (States) and No BR and yields nearly no correct explanations.

## 4.2   EVALUATING FUTURE ACTION PREDICTION

We further evaluate how well the LLM is actually able to reason over and explain current agent behavior by analyzing how well it can predict *future* agent behavior. Intuitively, if the LLM can accurately produce an explanation which identifies an agent's intent, then it should be able to use this intent to infer future actions. We evaluate this by issuing a follow-up prompt to the LLM for each explanation produced in the previous analysis to predict the agent's next action while reasoning with the explanation it produced.

**Explanations produced with BR (Path) enable accurate action prediction.** Tables 2 and 3 show that the LLM is able to effectively predict the agent's next actions when reasoning with explanations produced by BR (Path), consistently yielding 80-90% accuracy across all behavior types. There is one exception to this: short-term goal action prediction which yields 40% accuracy. This is due to the *locality* of the decision path – the path only encodes decision rules relevant to the agent's current action, which is highly correlated with the agent's current goal. If that goal happens to be short-term, meaning it will be achieved with the agent's *current* action, then the decision path rarely encodes enough information to reason about the *next* action. We conjecture that providing additional information, such as the current observation, in addition to the decision path behavior representation, can alleviate this issue. This is also the cause for the low action prediction accuracy over *Fixed* states in Table 3; the agent's strategy is often successfully identified, but the LLM exhibits uncertainty due to the lack of information contained within the decision path.

**Predictions can be right for the wrong reasons.** The BR (States) and No BR methods perform worse in action prediction accuracy and approximately align with explanation accuracy, indicating that in most cases, it is difficult to predict future behavior if the agent's decision-making rationale cannot be identified. However, there is an exception to this which is the relatively high accuracy of 60% for BR (States) when predicting over *Fixed* policy states (Table 3). On analysis, we found that the LLM can identify the simple action distribution produced by the pre-determined path (the agent moves in a north-south pattern) from the set of provided state-action samples, which is further narrowed down by spatial reasoning constraints, e.g., the agent can't move further north if it is already in the northern-most row. However, the LLM is unable to reason about *why* the agent

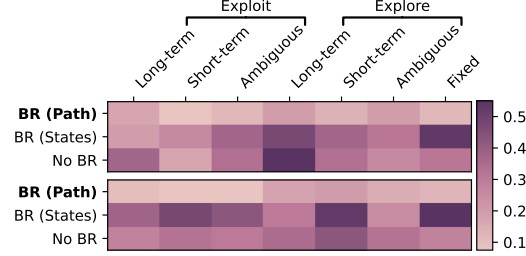

| | Method | Strategy | Action | Intent |
|---|---|---|---|---|
| **Fixed** | **BR (Path)** | **0.80** | 0.40 | **0.25** |
| | BR (States) | 0.05 | **0.65** | 0.00 |
| | No BR | 0.00 | 0.35 | 0.00 |

Table 3 & Figure 4: Left: Explanation and action prediction metrics for the *Fixed* policy. Right: hallucination rates for each method in generated explanations (top) and action predictions (bottom).

follows such an action distribution, leading to a case where actions are predicted correctly but the agent's rationale is not.

### 4.3 EVALUATING HALLUCINATION

The frequency of hallucination in the explanations and action predictions is shown in Fig. 4.

**Hallucination is significantly reduced with BR (Path)**. There are two interesting insights from the hallucination evaluation: a) explanations produced with the decision path representation yield far lower hallucination rates across all categories than the other methods, and b) BR (States) sometimes yields *more* hallucinations than No BR. This is a counterintuitive result, but we find that when no behavior representation is provided to the LLM, it tends to make conservative predictions resulting in fewer hallucinations at the cost of lower explanation and accuracy prediction accuracy.

**Hallucination is not correlated with action prediction accuracy**. Intuitively, we might think that the hallucination rate of the generated explanations is inversely correlated with the action prediction accuracy. That is, hallucinations are symptomatic of LLM uncertainty regarding agent intent and inject additional errors into the downstream reasoning process. However, we find this not to be the case and find no significant correlations between hallucination and action prediction metrics according to the Pearson correlation coefficient with $p < 0.05$.

## 5 PARTICIPANT STUDY AND ANALYSIS

While the quantitative analysis indicates that the explanations produced with our proposed behavior representation are accurate, we seek to answer whether the explanations are useful to humans. To answer this question we conduct two IRB-approved user studies with the following hypotheses:

**H1**: *Participants prefer the explanations produced by our method – BR (Path) – over the explanations produced by both BR (States) and a textual representation of the decision path (Template).*

**H2**: *Participants will not prefer the explanations produced by a human domain expert over ours.*

**H3**: *Participants will find follow-up interaction helpful for understanding the agent's behavior.*

### 5.1 EVALUATING EXPLANATION HELPFULNESS

The first study is designed to determine whether human participants find our explanations helpful in understanding an agent's behavior. We followed a within-subjects design where we presented each participant with a state-action pair, a visualization of the world state, and a pair of explanations. Each participant is asked to choose whether they find the first or second explanation more helpful in understanding the agent's behavior or whether they are equally helpful. This study considers two additional baselines in addition to BR (Path) and BR (States): **Human)** Natural language explanations produced by a domain expert with full knowledge of the agent's behavior; and **Template)** An algorithmic translation of the decision path to a textual representation. The Human explanations are intended to serve as an upper-bound on explanation quality, and the Template explanations a lower-bound. We recruited 40 participants who collectively answered 1106 questions, with the results shown in Fig. 5 (left). From these results, we find that hypothesis **H1 is fully supported**,

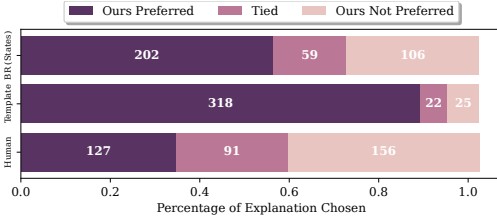 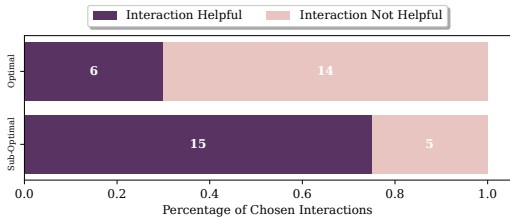

Figure 5: Left: Participant preference when presented with two explanations and asked to choose which is most helpful to understand agent behavior: Ours vs. BR (States) (top), Ours vs. Template (middle), and Ours vs. Human (bottom) over 1106 responses from 40 participants. Right: Helpfulness of interaction after being presented with an explanation with respect to *Explore* and *Exploit* policies across 40 responses from 10 participants. In both cases, Ours refers to BR (Path).

as computed with a one-tailed binomial test with $p < 0.05$. Participants significantly prefer the explanations produced by BR (Path) over both those produced by BR (States) as well as Template. We find that hypothesis **H2 is supported** as well, and participants did *not* prefer the explanations produced by human domain experts over those generated by our proposed method.

**Participants' preference is not influenced by hallucination.** Furthermore, we note that participant choice was not influenced by hallucinations present within explanations. Half of the explanations produced by BR (States) had some form of hallucinated fact or reasoning. However, we find no significant difference in participant preference when comparing the subset of explanations with hallucination against the subset of explanations without hallucination. This indicates that either a) hallucinated facts do not diminish explanation helpfulness in the eyes of participants, or b) participants fail to identify hallucinated information. As a result of our study, we observe a similar preference between BR(Path) vs. BR(States, No-Hallucination) and BR(Path) vs. BR(States, Hallucination).

## 5.2 EVALUATING EXPLANATION INTERACTIONS

We next evaluated whether participants found the ability to interact with the LLM and the generated explanations helpful. We performed a within-subjects study where 10 participants were recruited and presented with a series of state-action pairs, natural language explanations, and an interactive chat window to an LLM. Participants were given a period of 5 minutes to interact with the system and then they were asked to indicate whether they found the ability to interact with the LLM helpful for understanding the agent's behavior, or whether the initial explanation was sufficient. The results over 40 total responses are shown in Fig. 5 (right). The results **partially support hypothesis H3** and led to an interesting observation: human participants often assumed the agent would act exploitatively, much like the LLM. When the agent acted according to an exploitative strategy its actions aligned with the participants' expectations, and interaction was *not* found helpful. However, when the agent acted according to an exploration strategy the action was unexpected, and participants found the ability to issue follow-up queries to the LLM helpful for understanding agent behavior. We found participant interactions largely fell into three categories: clarification questions, counterfactual questions, and requests for concision. An example of such an interaction is shown in Fig. 3.

## 6 CONCLUSION AND FUTURE WORK

In this work, we propose a model-agnostic framework for producing natural language explanations for an agent's behavior. Through construction of a *behavior representation*, we are able to prompt an LLM to reason about agent behavior in a way that produces plausible and useful explanations, enables a user to interact and issue follow-up queries, and results in a minimal number of hallucinations, as measured through two participant studies and empirical experiments. While we recognize that our proposed method has limitations, namely that it requires distillation of an agent's policy into a decision tree which only works with non-dense inputs, we feel this is a promising direction for explainable policies. Such limitations can be overcome with more complex behavior representations, e.g., differentiable decision trees or concept feature extractors, and we expect the quality of explanations to improve as LLMs grow more capable.

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

# A  APPENDIX

To ensure the reproducibility of this study, the prompts used for querying the large language model are provided. These prompts contain four segments: a) a concise description of the environment the agent is operating in, e.g., state and action descriptions, b) a description of what information the behavior representation conveys, c) in-context learning examples, and d) the behavior representation and action that we wish to explain. While the final part is produced by the surrogate model on a case-by-case basis, the first three parts are supplied for reference.

**Environment Description**

Within the environment description segment, the Large Language Model (LLM) is acquainted with the game rules to orient it within the USAR settings, aiming to counter any misconceptions spurred by the LLM's general knowledge. Through testing, it is determined that GPT-4 displays considerable spatial reasoning skills. However, its north/south direction is opposite to our configurations, requiring explicit clarification. Additionally, we also address the contradiction between LLM knowledge and domain knowledge in this section. For instance, while the LLM assumes that rubble block the movement of agents, it does not in our context. The domain knowledge prompt is shared below:

The game is a search and rescue game set in a grid world. There are two agents: a medic and an engineer. The grid is made up of rooms, each represented by coordinates (x,y), where x represents the east-west direction, and y represents the north-south direction. Specifically, a larger x value corresponds to a location further east, and a larger y value corresponds to a location further south, with y=0 being the northernmost row and increasing y values moving southward.

- Both the engineer and the medic can move to an adjacent room in any of the four cardinal directions (north, south, east, west) during a single move.

- The medic has the ability to rescue a victim during a single move; however, this action cannot be performed concurrently with movement to an adjacent room.

- The engineer has the ability to remove rubble during a single move; this action also cannot be performed concurrently with movement to an adjacent room.

- Both agents can only perceive what's inside their current grid but have memory and can communicate instantly. Every grid visited by either of the agents is visible to both.

- Rooms may initially be unexplored, in which case victims or rubble are assumed to be non-existent. Once explored, details about victims and rubble will be updated.

- Rubble may or may not hide a victim. If a room contains rubble, victim is assumed to be non-existent. Removing the rubble may expose a hidden victim.

- Rubble in the room won't affect the movement of the agents.

**Behavior Representation Description**

A description of the behavior representation is provided to the LLM such that it knows how to reason with the information we provide. We emphasize that the LLM should **interpret the policy of a neural network rather than formulating its own rationale**. It should adhere to the behavior representation provided to accurately elucidate the neural network's policy. The respective task description prompt reads as follows:

Given an observation of the environment, the agent chooses to look at a subset of features in order to choose its action, which we denote as "Features". These features represent parts of the observation that are directly related to why the agent chose its action. Given this subset of features that the agent looks at, provide a concise explanation for why the agent chose the action that it did. Keep in mind that the agent may not always be making optimal

decisions, so consider that possibility if there seems like no reasonable goal the agent could be trying to accomplish from the set of features.

**In-Context Learning (ICL) examples**

The In-Context Learning (ICL) examples supply the LLM with instances of the expected outputs when an observation & action pair is presented. In this segment, we provide examples written by human domain experts. Without prior knowledge of the policy's type, human is given the same description as above and try to formulate based on the provided feature & action pair. We incorporated three samples, one each from the three possible behaviors *Explore*, *Exploit*, *Fixed*. Below is the human-written example for the *Explore* policy.

Features:
room (0, 1) doesn't contain rubble.
engineer is not in room (0, 1).
room (3, 2) doesn't contain rubble.
engineer is not in room (3, 2).
room (3, 3) doesn't contain rubble.
engineer is not in room (3, 3).
room (3, 4) contains rubble.
engineer is not in room (3, 4).
engineer is not in room (1, 1).
engineer is not in room (3, 0).
room (1, 3) doesn't contain rubble.
engineer is not in room (1, 3).
engineer is not in room (0, 4).
room (0, 4) doesn't contain rubble.
room (3, 1) doesn't contain rubble.
engineer is not in room (3, 1).
room (2, 4) doesn't contain rubble.
engineer is not in room (2, 4).
medic is not in room (2, 4).
medic is not in room (2, 1).
engineer is not in room (1, 2).
room (2, 3) doesn't contain a victim.
medic is not in room (3, 2).
medic is not in room (3, 1).
room (2, 3) doesn't contain rubble.
engineer is not in room (2, 3).
room (1, 4) doesn't contain rubble.
engineer is not in room (1, 4).
room (0, 3) has been explored.
room (0, 3) doesn't contain a victim.
medic is not in room (1, 4).
room (2, 2) contains rubble.

Action taken by the engineer:

engineer moves east to room (1, 2).

Explanation:

The engineer moves east to room (1, 2) because the engineer is currently in room (0, 2) and room (2, 2) contains rubble. From the set of features it appears the engineer is primarily looking for rooms containing rubble so that it can remove them, and since room (2, 2) contains rubble the engineer may be moving east to reach room (2, 2) and remove the rubble.

