# OpenReview forum: "Understanding Your Agent: Leveraging Large Language Models for Behavior Explanation"
_ICLR.cc/2024/Conference — Submitted to ICLR 2024_

### Official Review · Reviewer_ccmh · 2023-10-31

**Soundness:** 3 good
**Presentation:** 3 good
**Contribution:** 1 poor
**Rating:** 3
**Confidence:** 4

**Summary:**

This work presents an approach by which a large language model (LLM) can generate natural language explanations of agent behavior based only on observations of the state and the agent's actions. The method involves first learning a decision tree representation of the agent's behavior and then feeding traces through this tree to an LLM, which provides an explanation of the user's behavior. Multiple approaches to scrutinize the quality of the generated explanations are used including determine the rate of hallucinations, determining whether the LLM understand the agent's behavior well enough to predict subsequent actions, and user studies that seek to determine whether a user prefers explanations generated via the proposed approach. Notably, the experiments also allow users to interrogate the LLM for clarification or more detail about the provided explanations, a unique and potentially powerful capability in this domain.

**Strengths:**

The paper is very well written and clear. The introduction is strong and clearly communicates the core ideas in the paper. The motivation behind the problem under scrutiny is also clear. There were only a few points in reading that I did not follow what was going on.

The overall execution of the methodology also seems sound, though there are some general comments I have about the depth of the experiments. The idea of having an explanation system that an end user can interact with is a good point, and a clear argument in favor of this approach and a unique capability in this domain.

**Weaknesses:**

There are a few issues with the paper that I believe should be addressed and questions about the broad applicability of the proposed approach.

**The approach seems somewhat limited in scope, requiring a considerable amount of work from a designer and a description of the world model.**
The authors claim that the proposed approach can "generate natural language explanations for an agent's behavior based only on observations of states and actions." While this is technically true, there are significant additional requirements for the proposed approach to succeed. First, the agent's policy must be summarized into a decision tree, requiring a potentially lengthy training phase and access to the environment. In addition, the LLM is provided a comprehensive description of the world model, which it seems to need in order to generate reasonable explanations. The authors should comment on this constraint and address the scenario in which one might use this pipeline, since it seems to need access to low level knowledge about the environment (in the form of a model) such that other forms of explanation are more appropriate.

**Being able to summarize behaviors into a decision tree.**
Behaviors need to lend themselves to fairly simple summarization, not just for the LLM, but also because the requirement that a behavior tree could be used to represent the robot's behavior is a fairly stringent constraint. In many ways it seems that the ability of the system to be summarized in terms of a behavior tree is the more difficult of the two tasks. However, there is no way to know in advance if the behavior tree is a good proxy for the agent's behavior. Moreover, there are no experiments in the work to judge how accurate the behavior trees are.

There are only three policies being explored, each of which has a fairly simple high-level description. The lack of a more complex policy for which the system must generate explanations is notable, as it is unclear how effectively the system would be able to describe more sophisticated behaviors.

**On the subject of 'optimality'**
There are multiple places in which "optimal behavior" is mentioned or discussed, yet there is not a clear reward function. Presumably, the goal is to reveal and assist all of the victims as quickly as possible, and this seems to also be what the participants expect as well, since this is likely why they view the "exploratory" behavior as suboptimal. Some clarification on this point would be helpful.


**Concerns about the user studies**
Finally, the user study results are somewhat unconvincing to me and I worry that the format of the study are fundamentally flawed (I may be wrong here! I welcome comments by the authors). The purpose of explanations is to help a (non-expert) human end-user to understand why an agent behaved a certain way. The user study instead asks which explanation is preferred between two examples; this is potentially problematic since there is no way to determine if the user actually understands the agent's behavior or if they merely prefer the prose of one explanation over another. ChatGPT is well known to make high quality prose and can produce statements that are linguistically coherent and convincing even if the statements themselves are false in some way.

It is for this reason that I am concerned by the conclusion that hallucinations do not seem to impact user preference: this result may suggest that the user is indeed not understanding the agent's behavior, preferring only the quality of the prose in the explanations over their substance.

Instead, the user studies would have been much more effective if the user were evaluated on their ability to demonstrate a comprehension of the agent's behavior, for example asking the user to predict the agent's next action (as the LLM was asked in the earlier studies). This could be compared against the performance of a user if they were given the same high-level description of the agent's policy readers of this paper are provided: e.g., "explore" vs "exploit" policies.

Relatedly, it is concerning that the rate of hallucinations is so high, with ~30% hallucinations for the proposed BR (Path) approach.


**Conclusion**
Overall, I think the paper presents some interesting ideas, though I am unsure if its exploration of some of the relevant weaknesses or issues associated with the broad applicability of the approach is deep enough at this time.


**Minor comments**
- Sec. 4: I believe that the "Explore" and "Exploit" policy descriptions are flipped. As written, the "Explore" policy targets immediately clearing rubble.
- I do not believe that an example of the "Template" response is included, yet is important for understanding the user studies. The authors should provide an example of the "Template" explanation.
- The connection to MARL (multi-agent reinforcement learning) is not very clear. The authors mention that the explanations are being applied to a MARL task, yet for all of the example scenarios, there does not appear to be any synergy between the two robots in the domains, as they generally seem to have policies that are independent of one another. I recommend that the authors provide additional justification about how the experiments show the capacity of the system to explain a multi-agent system or remove the mention of the MARL scenario.
- While the colors in Figure 4 are useful at a glance, numbers should also be included. If the authors could add small-text numbers inside each of the corresponding boxes (keeping the colored background for each square), it would be helpful for understanding.

**Questions:**

Here I summarize the most important questions from my longer discussion above:

- The authors should comment on this constraint and address the scenario in which one might use this pipeline, since it seems to need access to low level knowledge about the environment (in the form of a model) such that other forms of explanation are more appropriate.
- Can the authors comment on the accuracy of the decision tree? Are there any experiments that seek to judge the rate with which the decision tree?
- The authors should comment on the utility of the user study and address potential issues with questions that seek only to judge preference between different forms of explanations. Notably, they should comment further on the lack of any impact of the high rate of hallucination on user preference.

---

> ### Author Response · Authors · 2023-11-21
> **Response to Reviewer ccmh (Part 1)**
>
> We would like to thank Reviewer ccmh for their time and detailed feedback. We appreciate that you find our motivation and methodology strong. Below is our response to your raised concerns and questions.
>
> **1) The approach seems somewhat limited in scope, requiring a considerable amount of work from a designer and a description of the world model**
>
> We argue that our approach actually requires considerably less effort than previous methods, which typically involve building a dataset of annotated natural language explanations for agent state-action pairs and fine-tuning a language model. Policy distillation is a fairly straightforward process (we use DAGGER in this work, see our response to #2 below) that only has to occur once. Similarly, a domain expert has to define a natural language description of the task MDP and a few in-context learning examples, but these are also only one-time efforts and are trivial in comparison to constructing an entire dataset for fine-tuning (as most prior methods require). We will clarify the writing to more clearly indicate that the description of the MDP is a requirement.
>
> **2) Being able to summarize behaviors into a decision tree**
>
> We use dataset aggregation (DAGGER)[1] to sample state-action pairs and train the decision tree (using CART). DAGGER proceeds by collecting a set of visited states at each iteration under the current distilled policy and the corresponding actions from the agent’s original policy, which is aggregated to the set of all previous state-action pairs and used to train the decision tree.
>
> The state of our environment is an NxMx5 tensor where N is the number of rows (5 in our case) and M is the number of columns (4). The last dimension represents a binary feature vector for each room and consists of: the presence of the medic, the presence of the engineer, the presence of rubble, the presence of a victim (may be hidden if there is rubble), and whether the room has been explored.
>
> [1] Ross, Stéphane, Geoffrey Gordon, and Drew Bagnell. "A reduction of imitation learning and structured prediction to no-regret online learning." AISTATS 2011.
>
> We quantified the distilled policy accuracy by rolling out 5000 episodes with random initializations, and computing the action accuracy using the original policy’s action as the target. The accuracy values are given below where we can observe a range from ~83% to 100%. We will add these results to the appendix of our paper.
>
> |             | explore | exploit | fixed   |
> |-------------|---------|---------|---------|
> | medic       | 87.72%  | 87.05%  | 100.00% |
> | engineer    | 82.94%  | 96.42%  | 100.00% |
>
> We would like to point out a few things with regard to the distilled policy.
> 1. Our environment was intentionally designed so as to exhibit ambiguous states, where different behaviors will produce the same action at time t, but different actions at time t+1. This ambiguity is the cause for some of the loss in performance (especially the explore policies).
> 2. We only produce explanations when the action of the distilled DT matches that of the original policy. So in the worst case here, we still explain ~83% of the agent’s actions, which we think is reasonable.

---

> ### Author Response · Authors · 2023-11-21
> **Response to Reviewer ccmh (Part 2)**
>
> **3) On the subject of ‘optimality’**
>
> We don’t assume optimal behavior, and will change the subtitle of Figure 5. The major confusion is likely raised by Figure 3, where the example includes the LLM response “The medic did not consider this information when making its decision, which would imply a non-optimal decision-making process on the part of the medic.” Note that this text is entirely generated by the LLM. Interestingly, GPT-4 assumes the exploitative policy as optimal and the explorative policy as suboptimal. Relevant discussions on this have been covered in section 4.1.
>
> **4) Regarding concerns about the user studies**
>
> The fact that humans didn’t identify explanations unfaithful to the agent doesn’t necessarily lead to the conclusion that our explanations didn’t convey sufficient information. Instead, it points out that users may overly pay attention to prose, so it’s particularly important to ensure the content provided to the users is faithful. To demonstrate this, we describe our user study in more depth. For each of the 20 cases, the user is given a figure of the current state, a figure of the future state (in which the agent accomplishes its next goal), and two explanations of random order. Then they are asked the question “which explanation best helps you understand the agent's behavior”. We provide the future state such that participants can better evaluate explanation helpfulness with respect to the agent’s actual behavior.
>
> In the study introduction, we explicitly informed participants that explanations may not be correct, so they were aware of this. However, the fact that participant preference was not affected by hallucination indicates either that, a) participants were unable to identify hallucinations even given the current and future states, or b) participants did identify it but considered it inconsequential. Our study design does not allow us to disentangle these effects, but it does not emphasize the necessity of using explanations which minimizes inaccuracies and hallucinations, which we show that our approach excels at.
>
> Due to our study design, we could not ask participants to predict the agent’s next action (because we provided them with the future state). In this case, we opted for giving more information to participants such that they can better judge explanation helpfulness by considering the agent’s future behavior, rather than withholding it and analyzing explanation comprehension by prompting participants to predict actions themselves.
>
>
> **5) Minor comments**
>
> As stated in the paper, Template refers to a textual representation of the decision path. Examples have been provided both in Figure 2 and in our appendix. Thanks for pointing out these issues, we’ll update the manuscript accordingly.
>
> Regarding the high hallucination rate, the maximum rate for BR (Path) is 20% for the explorative short-term policy, with the average rate falling at 10.7% across all policies. This is significantly lower than 30%, and we note that a considerable amount of hallucinations are from reasoning instead of from facts. Examples of them are “If the engineer is in room (0, 0) and moves north, they will be in room (0, 0)” while the engineer cannot move north in the first place, “there is no rubble blocking the way” while rubble is not blocking the way in our settings. Wrong presentation of facts is extremely rare in our proposed method.

---

> ### Comment · Reviewer_ccmh · 2023-12-04
> **Post-Rebuttal: Author comments helpful and improve clarity of approach, yet limitations with the method remain**
>
> I thank the authors for their detailed rebuttal comments; they were indeed helpful for clarity. Unfortunately, I share some of the concerns with the other reviews and also agree that the rebuttal comments (while helpful for elucidating the approach the authors have taken) do not resolve the more fundamental issues with the approach that I (and others) have pointed out. The two central lingering concerns are:
>
> 1. The simplicity of the proposed domain and policies. While having a simple domain is great for showing that the method aligns with designer expectations, it is insufficient to show its potential more broadly and I am not convinced that this will work on domains more complex than the one shown. I am also still a bit on the fence about whether the effort put forth by the designer is an issue. My instinct still remains that the designer's work to design; however, the authors' comments about this step being a "one time effort" did reduce my concerns somewhat.
> 2. The limitations of the user study, both in its lack of ability to test user's ability to use explanations to demonstrate better understanding of the agents and also the uncertainty over the impact of hallucinations (whether factual or not) are not addressed.
>
> I think there is potential for this work, and I am excited to see it move forwards. However, I do not think that some of the fundamental questions have been worked out yet. As such, my score remains unchanged.

---

### Official Review · Reviewer_Pb35 · 2023-10-31

**Soundness:** 3 good
**Presentation:** 2 fair
**Contribution:** 2 fair
**Rating:** 5
**Confidence:** 4

**Summary:**

This paper introduces a method for generating natural language explanations of an agent's behavior using large language models (LLMs) with a decision-tree representation of the agent's decision-making process. Given the ability to simulate a black-box agent policy, the method first distills a decision tree policy that is a surrogate of the original policy. To generate an explanation for why an agent took a particular action $a$ in state $s$, this surrogate policy is applied to the current state $s$, and a corresponding decision path (which the authors call a "behavior representation") is extracted from the decision tree. This decision path is then used to prompt LLM, which generates a human interpretable English representation of why the agent made that particular decision.

The authors demonstrate this method in the context of a search-and-rescue gridworld environment with partial observability. They distill decision-tree policies that approximate several different types of agent behavior, e.g. a policy that first explores all unexplored rooms before rescuing victims. They then apply their method to extract decision paths for observed state-action pairs, and to generate natural language explanations from those decision paths. To evaluate these explanations, the authors manually annotate the accuracy of the explanations with respect to several criteria (e.g. whether it accurately describes the agent's goal, or the agent's next action), and compare it several baselines, finding that explanations generated from decision paths are more accurate and less hallucinatory than explanations generated by baseline methods (e.g. when no decision path is provided to the LLM). The authors also perform a user study, finding that users prefer the explanations generated according to their method compared to the baselines, and that human-written explanations are not significantly preferred by users compared to the explanations generated by their method.

**Strengths:**

This paper attempts to develop a model-agnostic method for generating natural language explanations of agent behavior, relying on interpretable policy distillation and LLMs to generate plausible reasoning traces and corresponding explanations for why an agent took a particular action in a particular state. Insofar as the choice of model architecture cannot be controlled, a model-agnostic approach to explanation generation is potentially widely useful and applicable. In addition, the proposed method limits the tendency of LLMs to confabulate false or misleading explanations by providing the LLM with a reasoning trace from a decision tree surrogate of the original policy, improving the chances that the generated explanation is actually faithful to the original policy. Compared to directly providing the trace of the decision tree as an explanation, LLM generated explanations are more interpretable, while also being interactive, avoiding the pitfalls of traditional approaches to explainable and interpretable AI. Empirically, these benefits are borne out by the higher accuracy metrics of the proposed method compared to baselines, and the favorable user study.

**Weaknesses:**

While the idea behind this paper is interesting, several aspects of the paper raise questions about (i) how model-agnostic the approach actually is (ii) how well the results will generalize to other domains and policies, and (iii) how explanation quality is evaluated.

Regarding the model agnostic nature of the approach, it seems to me that a major requirement for the proposed method to work is that the representation of the environment itself is interpretable. Otherwise, there's no obvious way that the decision tree policy, and corresponding decision path, can be translated into an interpretable natural language template for the LLM to consume. While this requirement is satisfied by the gridworld environment evaluated by the authors, it is not true for a lot of other models and agent policies, which may directly consume pixels or low-level state representations as input. As such, one of the promised benefits of the proposed method --- that all it requires is simulation access to some policy --- is not really delivered upon. Rather, this method can only be applied if the underlying state representation is sufficiently interpretable. While that is a reasonable assumption to make, it should be clearly stated in the framing of the paper. And if such a framing is taken, then the method should ideally also be compared against XAI approaches that do leverage the assumption of interpretable state representations, such as plan recognition and goal recognition algorithms.

As for the whether the results will generalize to other environments and policies, one worry I have about the current set of experiments is that they seem highly tailored to generating explanations for either explorative policies or exploitative policies in the search and rescue setting. While the natural language explanations shown in the paper seem intuitively plausible, there's a much larger space of policies that purely exploratory or exploitative policies, and it's not obvious to me that either humans or LLMs would succeed in coming up with compelling natural language explanations of those policies. Even looking at the example decision paths / behavior representations shown in the paper, it seems to me like a wide variety of goals, strategies, and intentions could be consistent with those decision paths, and I worry that the fact that LLMs are generating the "right" explanations is just an artifact of the few-shot examples combined with the design of the test set. For example, would the LLMs still generate the "right" explanations if the test policies happened to be exploratory in the top half of the grid, and exploitative in the bottom half of the grid? Or if the true policies applied some mixture of exploration and exploitation, e.g. via Thompson sampling?

All of this is tied to questions I have about what counts as "good" explanation of an agent's behavior, and how that can be fairly evaluated. At least when it comes to explaining *human* behavior, we typically default to explaining others' actions in terms of mental states like goals, beliefs, intentions, etc, as part of our theory-of-mind. The authors seem to implicitly have something like this view in mind, given that they manually annotated the LLM-generated explanations for accuracy in terms of goals, intents, strategies, and other supposed mental states. However, it's not clear to me that this is the right evaluation metric for explanations about *artificial* agents, given that black-box policies could act very different from humans, and may not have anything like goals or intents at all! (In fact, a decision-tree policy is a purely reactive policy, and doesn't involve any goal-directed planning -- so there's a certain sense in which it's a metaphysical error to impute a goal to a reactive agent, unless the reactive policy was trained to imitate some goal-directed policy computed via value iteration.)

To avoid this anthropomorphic bias in explanation generation, a different standard for a "good" explanation is that good explanations should support both accurate prediction, but also accurate intervention and counterfactual reasoning. For example, one reason why we might attribute mental states like goals and beliefs to other humans is because they are useful abstractions that we can plan to intervene upon (see [1] for an exposition of this idea). In context of explaining the behavior of artificial agents, this could be evaluated by considering the accuracy of using the explanations for future action prediction (which the authors do), but also by seeing whether they support useful interventions. I think adding such evaluations would be a helpful improvement to the current paper. Otherwise, I worry that the current evaluation methodology is overfit to the particular policies considered by the authors, which admit relatively simple anthropomorphic explanations involving exploratory or exploitative goals.

[1] Ho, M. K., Saxe, R., & Cushman, F. (2022). Planning with theory of mind. Trends in Cognitive Sciences, 26(11), 959-971.

**Questions:**

- How model-agnostic is this approach, given that it seems to rely on interpretable state representations?

- Can we expect this method to generalize to wider a range of policies, or to a richer range of environments with much more features?

- How were the generated explanations hand-annotated, and what steps were taken to make sure that the evaluation/coding criteria weren't overfit to the choice of test policies?

---

> ### Author Response · Authors · 2023-11-21
> **Response to Reviewer Pb35 (Part 1)**
>
> We would like to thank Reviewer Pb35 for their thorough and thoughtful review. We appreciate the positive comments about our method and that you think our idea is interesting. We respond to your questions and concerns below.
>
> **1) How model-agnostic is this approach given that it seems to rely on interpretable state representations?**
>
> You are correct that the distillation process used here requires inputs which can be directly consumed by a decision tree (and also interpreted by an LLM, but the DT is the bigger constraint in our experience). However, we also feel that our approach is applicable to environments which contain different state spaces, e.g. images or uninterpretable latent representations. We argue that such representations can be handled by first extracting a set of semantic features, e.g. visual concepts [1], which are then used to distill a structured behavior representation. Ultimately this is an issue that applies to all interpretable/explainable methods – how do we construct human-intelligible representations from inherently uninterpretable ones – and so we don’t feel that this is unique to our method, particularly as we are not directly addressing this problem.
>
> [1] Koh, Pang Wei, et al. "Concept bottleneck models." ICML 2020.
>
> **2) Can this method generalize to a wider range of policies, or to a richer range of environments?**
>
> These are reasonable points, which we think can be reduced to: how descriptive can a decision tree/path be of an agent’s strategy, intent, and goal? We agree that it may be possible for a given decision path to be interpreted multiple ways, and that the generated explanation is influenced by the in-context learning examples which are given in the prompt. However, we feel that is also a strength of this approach: if an agent is capable of producing similar-looking decision paths that may have different underlying intents, then this can be represented through in-context learning examples which themselves express contradictory explanations or uncertainty. We noted this behavior ourselves in the paper, e.g. when providing the model with no behavior representation (no DT path), the LLM often exhibited uncertainty and refrained from generating specific explanations for agent behavior (even when using the “standard” set of in-context examples in this case). There is even a potential for prompting the LLM for a set of explanations and then measuring agreement above them to quantify certainty.
>
> Despite this, we emphasize that our approach has several benefits over previous methods:
> 1. We leverage the encoded knowledge within the pre-trained LLM to avoid having to curate annotated datasets of states and explanations, which is impractical when attempting to explain arbitrary agent behavior.
> 2. The LLM itself is capable of performing reasoning over the given inputs (guided by the in-context examples), which often leads to more useful explanations as a human observer no longer has to perform this reasoning themselves (e.g. from textual translations of decision paths).
>
> Lastly, we want to note that decision trees are rather expressive in terms of behavior. For example, with your example of an agent that is exploratory in the top-half of the grid and exploitative in the bottom-half, we would expect a decision tree to represent this with different decision path structures for rooms in the top half vs bottom half.

---

> > ### Author Response · Authors · 2023-11-21
> > **Response to Reviewer Pb35 (Part 3)**
> >
> > **4) Concerns over user study evaluation and interventions/counterfactuals**
> >
> > These are good points, and we would like to provide additional details for our user study here. When evaluating explanation helpfulness (Sec. 5.1) we showed two state images to the participant: the current state in which the agent takes the current action that we wish to explain, and the future state that occurs after the agent achieves its next goal (see discussion on goal-directed behavior above). This enables each participant to judge the explanations not only on their comprehensibility, but also on whether it is specifically helpful in understanding the agent’s actual behavior.
> >
> > As you pointed out, we did indeed evaluate action prediction accuracy. However, we argue that we also evaluated intervention/counterfactual accuracy via our second user study in Sec. 5.2. In this study, participants were presented with an agent state/action/explanation and offered the chance to interact with the LLM to issue clarification, counterfactual, or other queries. As we observed, participants found this interaction helpful when the agent’s actions did not align with the participant’s expectations. We will add examples of these interactions to the appendix and we list an example below.
> >
> > *LLM Response 1 (Explanation)*
> >
> > `The engineer moves east to room (1, 0) because this room contains rubble. Based on the features, the engineer appears to prioritize removing rubble from rooms. Since the engineer is in room (0, 0), moving east to room (1, 0) aligns with this goal, demonstrating the engineer's focus on clearing rubble as part of the search and rescue operation. The other mentioned rooms do not contain rubble, making room (1, 0) the logical choice for the engineer's next action.`
> >
> > *Counterfactual Query*
> >
> > `What would the engineer do if room (0,0) also contains rubble? Answer briefly.`
> >
> > *LLM Response 2*
> >
> > `If room (0,0) also contains rubble, the engineer would likely stay in room (0,0) and take action to remove the rubble there, given their priority of clearing rubble in the search and rescue operation.`

---

> ### Author Response · Authors · 2023-11-21
> **Response to Reviewer Pb35 (Part 2)**
>
> **3) How were the examples hand-annotated and how did we avoid over-fitting to to our choice of policies?**
>
> We first note that the agent policies themselves were exhibiting goal-directed behavior in our experiments by design (we explicitly crafted policies to exhibit exploitation and exploration type behaviors). Any reasonably accurate distilled policy will then mimic such goal-directed behavior as well. This actually ties into our broader point: we don’t necessarily agree that goal-directed metrics are inappropriate for explaining agent behavior in general. In some sense, any explanation which attempts to reason about high-level behavior is at risk of “anthropomorphic bias” unless a policy explicitly includes hierarchical behavior planning or computational theory of mind models. And yet, we can clearly observe goal-directed behavior in standard deep neural network policies that do not include such mechanisms [2] – to such a degree that it is useful for humans to reason about agent behavior in terms of them. So while there may be cases in which inferred goals do not explicitly represent the underlying policy, we feel that the idea and metrics in general still have merit, particularly from a comprehensibility standpoint for humans.
>
> [2] Vinyals, Oriol, et al. "Grandmaster level in StarCraft II using multi-agent reinforcement learning." Nature 2019.
>
> We include additional details regarding the hand-annotation below.
>
> We designed *policy-agnostic quantitative metrics* to make sure our evaluation approach can be applied to other tasks and domains. These are Strategy, Category, Goal, Action, and Intent accuracy. These metrics are intended to work with any environment and agent that exhibits high-level strategies and low-level goals so as to be useful for future work. The downside is that such metrics must be manually annotated from natural language explanations, requiring considerable human effort. However, we find these provide far more detailed insights than automated metrics which may be erroneous (e.g. asking an LLM to evaluate itself) or only operate at the dataset-level (e.g. BLEU, BERTscore). We define our metrics in the manuscript in Sec. 4 but restate them here with additional details.
>
> Suppose that the engineer is taking an exploitative strategy, and the decision path largely consists of decision rules operating over rubble locations (similar to the same path in Fig. 2).
>
> • Strategy: Whether the overall strategy is identified. In this case, whether the explanation contains “The agent seems to prioritize the removal of rubble” for an exploitative engineer, as opposed to “The agent seems to be prioritizing exploration” for an explorative engineer.
>
> • Category: Whether the agent’s goal category was identified in the explanation. For example, if the explanation is “The engineer is prioritizing the removal of rubble and moves south to room (0,1) as that brings it closer to room (0,4) which contains rubble”, then the explanation correctly identifies that the agent is seeking out rubble (which happens to be in room (0,4) in this case, although the specific location is not part of the category).
>
> • Goal: Whether the agent’s specific goal was correctly identified in the explanation. In the explanation above, if the agent’s actual goal was to remove the rubble in room (0,4) (which we can observe by looking at future actions), then the goal is correct. However, if the agent instead moved south to room (0,1) in order to then move east and remove rubble in room (1,1), the goal would be incorrect even though the category would still be correct.
>
> • Action: Whether the agent’s next action was successfully predicted.
>
> • Intent: Whether the agent’s intent for taking the next action was successfully identified. If the explanation is “The engineer will move south to room (0, 3) to get closer to room (0, 4) which contains rubble” for an exploitative policy, Intent is correct. If the explanation is “The engineer will move south to room (0, 3) to get closer to room (1, 4) which is unexplored” for an exploitative policy (which should take rubble in room (0, 4) as priority), Intent is incorrect.

---

### Official Review · Reviewer_NiyW · 2023-11-01

**Soundness:** 2 fair
**Presentation:** 2 fair
**Contribution:** 2 fair
**Rating:** 5
**Confidence:** 4

**Summary:**

In this paper, the author proposed an model agnostic approach to generate natural language explanations for an agent's behavior based only on observations of states and actions, To be specific, it's implemented through a 3 step approach: first distill the agent's policy into a decision tree.  Second generate a behavior representation from the decision tree. Third query an LLM for an explanation given the behavior representation.  The proposed method is evaluated in a multi-agent search-and-rescue environment and showed the effectiveness of the proposed method.

**Strengths:**

The paper proposed a novel framework to generate natural language explanations of an agent's behavior given only observations of states and actions. By make such assumption, the proposed method is model agnostic for both the agent's policy model and LLM, which makes it pretty independent.

**Weaknesses:**

The proposed method is supposed to be a general framework, however, in the paper, it's only verified in one domain with limited data. It's better to do more experiments on more data to prove the effectiveness of the proposed method.

**Questions:**

1. For the experiment setup section, can you explain more on the metrics, for example, what's the difference between Category and Goal?
2. Can you provide more details on the decision tree based distillation, such as what features used?
3. For the distilled decision tree, how does the its performance compared with original policy model? how will this performance gap influence the generated explanation?
4. As proposed in the conclusion, this method is limited  by using only non-dense inputs, however, for real world application, many of them will include dense inputs like continuous features, this makes the proposed method limited.

---

> ### Author Response · Authors · 2023-11-21
> **Response to Reviewer NiyW (Part 1)**
>
> We would like to thank Reviewer NiyW for their time and helpful feedback. We appreciate that you consider the model-agnostic nature of our method a strength. Below is our response to your raised concerns and questions.
>
> **1) Additional information on metrics**
>
> We designed *policy-agnostic quantitative metrics* to make sure our evaluation approach can be applied to other tasks and domains. These are Strategy, Category, Goal, Action, and Intent accuracy. These metrics are intended to work with any environment and agent that exhibits high-level strategies and low-level goals so as to be useful for future work. The downside is that such metrics must be manually annotated from natural language explanations, requiring considerable human effort. However, we find these provide far more detailed insights than automated metrics which may be erroneous (e.g. asking an LLM to evaluate itself) or only operate at the dataset-level (e.g. BLEU, BERTscore). We define our metrics in the manuscript in Sec. 4 but restate them here with additional details.
>
> Suppose that the engineer is taking an exploitative strategy, and the decision path largely consists of decision rules operating over rubble locations (similar to the same path in Fig. 2).
>
> • Strategy: Whether the overall strategy is identified. In this case, whether the explanation contains “The agent seems to prioritize the removal of rubble” for an exploitative engineer, as opposed to “The agent seems to be prioritizing exploration” for an explorative engineer.
>
> • Category: Whether the agent’s goal category was identified in the explanation. For example, if the explanation is “The engineer is prioritizing the removal of rubble and moves south to room (0,1) as that brings it closer to room (0,4) which contains rubble”, then the explanation correctly identifies that the agent is seeking out rubble (which happens to be in room (0,4) in this case, although the specific location is not part of the category).
>
> • Goal: Whether the agent’s specific goal was correctly identified in the explanation. In the explanation above, if the agent’s actual goal was to remove the rubble in room (0,4) (which we can observe by looking at future actions), then the goal is correct. However, if the agent instead moved south to room (0,1) in order to then move east and remove rubble in room (1,1), the goal would be incorrect even though the category would still be correct.
>
> • Action: Whether the agent’s next action was successfully predicted.
>
> • Intent: Whether the agent’s intent for taking the next action was successfully identified. If the explanation is “The engineer will move south to room (0, 3) to get closer to room (0, 4) which contains rubble.” for an exploitative policy, Intent is correct. If the explanation is “The engineer will move south to room (0, 3) to get closer to room (1, 4) which is unexplored.” for an exploitative policy (which should take rubble in room (0, 4) as priority), Intent is incorrect.
>
> **2) More details on tree-based distillation**
>
> We use dataset aggregation (DAGGER)[1] to sample state-action pairs and train the decision tree (using CART). DAGGER proceeds by collecting a set of visited states at each iteration under the current distilled policy and the corresponding actions from the agent’s original policy, which is aggregated to the set of all previous state-action pairs and used to train the decision tree.
>
> The state of our environment is an NxMx5 tensor where N is the number of rows (5 in our case) and M is the number of columns (4). The last dimension represents a binary feature vector for each room and consists of: the presence of the medic, the presence of the engineer, the presence of rubble, the presence of a victim (may be hidden if there is rubble), and whether the room has been explored.
>
> [1] Ross, Stéphane, Geoffrey Gordon, and Drew Bagnell. "A reduction of imitation learning and structured prediction to no-regret online learning." AISTATS 2011.

---

> ### Author Response · Authors · 2023-11-21
> **Response to Reviewer NiyW (Part 2)**
>
> **3) How the performance of the tree compares with the surrogate policy, and how the performance gap influence the generated explanations**
>
> We quantified the distilled policy accuracy by rolling out 5000 episodes with random initializations, and computing the action accuracy using the original policy’s action as the target. The accuracy values are given below where we can observe a range from ~83% to 100%.
>
> |             | explore | exploit | fixed   |
> |-------------|---------|---------|---------|
> | medic       | 87.72%  | 87.05%  | 100.00% |
> | engineer    | 82.94%  | 96.42%  | 100.00% |
>
>
> We would like to point out a few things with regard to the distilled policy.
> 1) Our environment was intentionally designed so as to exhibit ambiguous states, where different behaviors will produce the same action at time t, but different actions at time t+1. This ambiguity is the cause for some of the loss in performance (especially the explore policies).
> 2) We only produce explanations when the action of the distilled DT matches that of the original policy. So in the worst case here, we still explain ~83% of the agent’s actions, which we think is reasonable.
>
>
>
> **4) Generalization to environments with continuous/dense inputs**
>
> Our approach readily supports continuous observations and action spaces as-is; decision trees can split over continuous features and also be used for regression tasks (continuous actions), and most tree-construction algorithms natively support this, e.g. CART, ID3. High-dimensional inputs such as images can also be supported by first extracting a set of semantic features from the images, e.g. visual concepts [1].
>
> As for complex environments which induce complex behaviors, this gets into the issue of how complex of a policy a decision tree can encode. While we do not directly investigate this question in our work, we can conjecture here about what we expect. Previous works [2] have shown that DTs can achieve surprisingly good performance purely through distillation, however, it is likely that this approach may not work for all complex policies. When that is the case, we expect that our method would also work with other model representations for the behavior. For example, instead of a decision tree maybe we utilize a gradient boosted tree or a differentiable decision tree (that is “crispified” [3]). While we have not investigated these specific approaches yet, LLMs have shown at least some ability to interpret even complex neural network representations [4], so we expect it would perform at least as well on decision tree variants.
>
> [1] Koh, Pang Wei, et al. "Concept bottleneck models." ICML 2020.
> [2] Bastani, Osbert, Yewen Pu, and Armando Solar-Lezama. "Verifiable reinforcement learning via policy extraction." NeurIPS 2018.
> [3] Paleja, Rohan, et al. "Learning Interpretable, High-Performing Policies for Autonomous Driving." RSS 2022.
> [4] Bills, Steven, et al. "Language models can explain neurons in language models." 2023.

---

### Official Review · Reviewer_5qZT · 2023-11-01

**Soundness:** 2 fair
**Presentation:** 3 good
**Contribution:** 2 fair
**Rating:** 3
**Confidence:** 2

**Summary:**

Authors propose a framework to explain the behavior of an agent. The framework consists of three phases, where in phase 1, the agent's policy is approximated by a decision tree. In the next phase, authors represent the decision paths of the decision tree as behavior representations. The last phase takes these behavior representations and feeds them to a language model using in-context learning, to produce a natural language explanation/output. Evaluation of the framework is done both quantitatively and qualitatively. In the quantitative evaluation, the quality of the behavior representation is assessed based on the reasoning performance and the future behavior prediction of the language model. Authors have conducted a human-subject study to evaluate the explanation quality. Further evaluations were reported on the hallucinations of the language model.

While the proposed framework presents an interesting methodology that can produce natural explanations, there are some major drawbacks and weaknesses which I detail below.

**Strengths:**

-The method is model agnostic, and can be used with any agent described by an MDP.
-The framework allows for interactive explanations, which enables the user to clear any doubts using back and forth dialog.
-Framework can be quickly adapted to existing agents, with limited computing resources and implementation overhead.
-Evaluation is done both quantitatively and qualitatively, which I see as a strength in an interpretability method.

**Weaknesses:**

There are major drawbacks of the method detailed below.

-The validity and the correctness of the behavior explanations are assessed by the language model using feature behavior inference,  instead of the human user (as the intended end user). This is generally assessed by action prediction (where we ask the human participant to predict the future behavior of the agent, given some previous explanations). Refer to Metrics for XAI of Hoffman et.al [1].
-There are several surrogate models that approximate the reasoning of the base agent, which can affect the faithfulness of the explanations. Ideally the faithfulness of the explanations should be more thoroughly evaluated.
-There are few details given of the agent, the decision tree, environment and the language model used. Authors have used one type of agent, in one environment and used one language model, which raises further questions of the methods generalizability and scalability to complex agents and larger environments.
- The quality of the explanation depends heavily on the task description, and this might be harder to craft in more complex environments.
- The human-subject experiments are only used to evaluate the preferences of the participants. Explanations can be highly preferred but be misleading (i.e. unfaithful to the underlying agent).

[1] https://arxiv.org/abs/1812.04608


---- after the rebuttal ----

I acknowledge the rebuttal of the authors. Some of my major concerns still remain.

**Questions:**

- Can this framework handle domains with continuous actions? If this is the case, what are the changes that the framework needs?
- The selection of the language model can have a major impact on the explanations, have authors tested the framework with any other language models?

---

> ### Author Response · Authors · 2023-11-21
> **Response to Reviewer 5qZT (Part 1)**
>
> We would like to thank Reviewer 5qZT for their time and insightful comments. We are glad that you appreciate the model-agnostic nature of our approach, the interactivity it enables, and our qualitative/quantitative evaluation. Below is our response to your raised concerns and questions.
>
> **1) Few details are given of the agent, the decision tree, environment, and the language model**
>
> We have provided details regarding our experiments in Appendix A, but we restate them here with some additional detail for clarity:
>
> **Environment**: The environment is a partially observable multi-agent urban search and rescue task, consisting of a grid of inter-connected rooms that may contain rubble and/or victims. There are two agents with heterogeneous skills: a medic which can save victims and an engineer which can remove rubble. The goal of the task is to save the victims, however, they may be trapped by rubble if both a victim and rubble are present in the same room. In this case, the agents must work together: the engineer must remove the rubble so the medic can save the victim.
>
> **Agent**: We leverage two types of agent behaviors, an exploration type and an exploitation type. The exploration type will first explore all rooms to identify exposed victims and rubble (which may possibly hide trapped victims). The exploitation type acts as a greedy agent, which begins by exploring but acts to either save victims or remove rubble (depending on the agent type) as soon as one is found. All of our results produce explanations for both the medic and engineer agents.
>
> **Language model**: We use GPT4 via OpenAI’s developer API. We include a concise description of the environment (similar to above), a description of the behavior representation (if there is one), and three in-context learning examples.
>
> **Decision tree**: We use dataset aggregation (DAGGER)[1] to sample state-action pairs and train the decision tree (using CART). DAGGER proceeds by collecting a set of visited states at each iteration under the current distilled policy and the corresponding actions from the agent’s original policy, which is aggregated to the set of all previous state-action pairs and used to train the decision tree.
>
> [1] Ross, Stéphane, Geoffrey Gordon, and Drew Bagnell. "A reduction of imitation learning and structured prediction to no-regret online learning." AISTATS 2011.

---

> ### Author Response · Authors · 2023-11-21
> **Response to Reviewer 5qZT (Part 2)**
>
> **2) Concerns about generalization to other environments, language models, and continuous action spaces**
>
> Our method was evaluated on a single environment due to the time-consuming nature of hand-annotating responses for quantitative evaluation and running user studies. However, we argue that the depth of our analysis on this environment offers intriguing insights, and this doesn’t limit the generalization of our approach to other environments.
>
> **Environment and policy-agnostic metrics**: We designed *policy-agnostic quantitative metrics* to make sure our evaluation approach can be applied to other tasks and domains. These are Strategy, Category, Goal, Action, and Intent accuracy. These metrics are intended to work with any environment and agent that exhibits high-level strategies and low-level goals so as to be useful for future work. The downside is that such metrics must be manually annotated from natural language explanations, requiring considerable human effort. However, we find these provide far more detailed insights than automated metrics which may be erroneous (e.g. asking an LLM to evaluate itself) or only operate at the dataset-level (e.g. BLEU, BERTscore). We define our metrics in the manuscript in Sec. 4 but restate them here with additional details.
>
> Suppose that the engineer is taking an exploitative strategy, and the decision path largely consists of decision rules operating over rubble locations (similar to the path in Fig. 2).
>
> • Strategy: Whether the overall strategy is identified. In this case, whether the explanation contains “The agent seems to prioritize the removal of rubble” for an exploitative engineer, as opposed to “The agent seems to be prioritizing exploration” for an explorative engineer.
>
> • Category: Whether the agent’s goal category was identified in the explanation. For example, if the explanation is “The engineer is prioritizing the removal of rubble and moves south to room (0,1) as that brings it closer to room (0,4) which contains rubble”, then the explanation correctly identifies that the agent is seeking out rubble (which happens to be in room (0,4) in this case, although the specific location is not part of the category).
>
> • Goal: Whether the agent’s specific goal was correctly identified in the explanation. In the explanation above, if the agent’s actual goal was to remove the rubble in room (0,4) (which we can observe by looking at future actions), then the goal is correct. However, if the agent instead moved south to room (0,1) in order to then move east and remove rubble in room (1,1), the goal would be incorrect even though the category would still be correct.
>
> • Action: Whether the agent’s next action was successfully predicted.
>
> • Intent: Whether the agent’s intent for taking the next action was successfully identified. If the explanation is “The engineer will move south to room (0, 3) to get closer to room (0, 4) which contains rubble.” for an exploitative policy, Intent is correct. If the explanation is “The engineer will move south to room (0, 3) to get closer to room (1, 4) which is unexplored.” for an exploitative policy (which should take rubble in room (0, 4) as priority), Intent is incorrect.
>
>
> **Environments with high-dimensional inputs, continuous action spaces, and higher complexity**: Our approach readily supports continuous action spaces as-is; decision trees can be used for regression tasks (continuous actions) and most tree-construction algorithms natively support this, e.g. CART, ID3. High-dimensional inputs such as images can also be supported by first extracting a set of semantic features from the images, e.g. visual concepts [2].
>
> As for complex environments which induce complex behaviors, this gets into the issue of how complex of a policy a decision tree can encode. While we do not directly investigate this question in our work, we can conjecture here about what we expect. Previous works [3] have shown that DTs can achieve surprisingly good performance purely through distillation, however, it is likely that this approach may not work for all complex policies. When that is the case, we expect that our method would also work with other model representations for the behavior. For example, instead of a decision tree maybe we utilize a gradient boosted tree or a differentiable decision tree (that is “crispified” [4]). While we have not investigated these specific approaches yet, LLMs have shown at least some ability to interpret even complex neural network representations [5], so we expect it would perform at least as well on decision tree variants.
>
> [2] Koh, Pang Wei, et al. "Concept bottleneck models." ICML 2020.
>
> [3] Bastani, Osbert, Yewen Pu, and Armando Solar-Lezama. "Verifiable reinforcement learning via policy extraction." NeurIPS 2018.
>
> [4] Paleja, Rohan, et al. "Learning Interpretable, High-Performing Policies for Autonomous Driving." RSS 2022.
>
> [5] Bills, Steven, et al. "Language models can explain neurons in language models." 2023.

---

> ### Author Response · Authors · 2023-11-21
> **Response to Reviewer 5qZT (Part 3)**
>
> **3) Preference is heavily involved in the user study regarding the evaluation of behavior explanations**
>
> The quality of the generated explanations are evaluated from a holistic perspective, with the quantitative metrics evaluating faithfulness and the human study evaluating preference. We believe that a good explanation needs to be both faithful and comprehensible from a human's perspective. It is true that users didn't prefer the non-hallucinated explanations over hallucinated ones (Sec. 5.1), but whether this is because they considered the hallucination inconsequential or because they didn’t detect it in the first place we cannot say. However, this supports our argument that we need to generate explanations which minimize hallucinations in the first place, and our quantitative results are intended to show that this is the case. If we use explanations with hallucinations and users blindly trust them, this can mislead the user and potentially lead to dangerous situations.

---

> > ### Comment · Reviewer_5qZT · 2023-11-22
> > **Response to the rebuttal**
> >
> > I thank the authors for clarifying comments.
> >
> > Regarding generalization to other environments, LLMs:
> >
> > I acknowledge the time consuming nature of handcrafting annotations and running user studies that authors have mentioned. My concern remains that the environment is not complex enough to argue about the strengths of the method's generalizability.
> >
> > Regarding the human study:
> >
> > My concern still remains on the faithfulness of explanations. Users need to be queried on their understanding of the agent to evaluate if they have understood the agent's behaviour through explanations (in addition to assessing the subjective quality of the explanations).
> >
> > I believe the manuscript can be greatly improved by addressing the above concerns.

---

### Meta-Review · Area_Chair_WFny · 2023-12-04

**Metareview:**

Reviewers see potential in this approach but it currently seems too early on and there are a some shared concerns than remain unaddressed.

Reviewers all found the domain narrow. While narrow domains are the way to start, the hope is that these allow for clean / convincing results. But reviewers also found the user study not as convincing (high rate of hallucination in model outputs, imperfect user discretion in what a good explanation is). Either expanding the domain or making the evaluation stronger /  more convincing on this simpler domain would be great directions to take this.

**Justification For Why Not Higher Score:**

Reviewers all found the domain narrow. While narrow domains are the way to start, the hope is that these allow for clean / convincing results. But reviewers also found the user study not as convincing (high rate of hallucination in model outputs, imperfect user discretion in what a good explanation is). Either expanding the domain or making the evaluation stronger /  more convincing on this simpler domain would be great directions to take this.

**Justification For Why Not Lower Score:**

NA

---

### Decision · Program_Chairs · 2024-01-16

Reject